# Distributionally Robust Surface Reconstruction from Sparse Point Clouds

## Abstract

We consider the problem of learning Signed Distance Functions (SDF) from sparse and noisy 3D point clouds. This task is significantly challenging when no ground-truth SDF supervision is available. Unlike recent approaches that rely on smoothness priors, our method, rooted in a distributionally robust optimization (DRO) framework, incorporates a regularization term that leverages samples from the uncertainty regions of the model to improve the learned SDFs. Thanks to tractable dual formulations, we show that this framework enables a stable and efficient optimization of SDFs in the absence of ground truth supervision. Through extensive experiments and evaluations, we illustrate the efficacy of our DRO inspired learning framework, highlighting its capacity to improve SDF learning with respect to baselines and the state-of-the-art using synthetic and real data evaluation.

## 1 Introduction

3D reconstruction from point clouds is a long standing problem at the intersection of computer vision, graphics and machine learning. While classical optimization methods such as Poisson Reconstruction (Kazhdan & Hoppe, 2013; Hou et al., 2022) or Moving Least Squares (Guennebaud & Gross, 2007) can be effective with dense, clean point sets and accurate normal pre-estimations, recent deep learning-based alternatives provide more robust predictions, particularly for noisy and sparse inputs, bypassing the need for normal data in many cases. In this regard, several existing methods rely on deep priors learned from large fully labeled 3D data such as the synthetic dataset ShapeNet (Chang et al., 2015). However, this strategy entails computationally expensive trainings, and the resulting models can still be prone to out-of-distribution generalization issues, as pointed by (Chen et al., 2023a; Ouasfi & Boukhayma, 2024c), whether caused by change in the input density or domain shift. As a matter of fact, Table 2 shows that our unsupervised method outperforms supervised generalizable models when testing on data that is sparser and different in nature from their training corpus. Therefore, it is important to design learning frameworks that can lead to robust reconstruction under such extreme constraints.

Recent work (Ouasfi & Boukhayma, 2024a) shows that strategies that can successfully recover SDF representations from dense point clouds such as Neural-Pull (NP) (Ma et al., 2021) often struggle when the point cloud is sparse and noisy due to overfitting. As a consequence, the extracted shapes have missing parts and hallucinations (*cf.* Figures 4,2). Instead or relying on smoothness priors, Ouasfi & Boukhayma (2024a) focus on how training distributions affect performance of the SDF network. They introduce distributionally robust optimization for sdf learning and rely on pointwise adversarial samples to regularize the learning process. Within the DRO (Volpi et al., 2018; Rahimian & Mehrotra, 2019) framework, the loss is minimized over the worst-case distribution within a neighborhood of the observed training data distribution. In this paper we show that this procedure can be generalized to hedge against different types of perturbations and provide more robustness to noise. To measure the distance between distributions, various metrics have been explored in DRO literature including f-divergence (Ben-Tal et al., 2013; Miyato et al., 2015; Namkoong & Duchi, 2016), alongside the Wasserstein distance (Blanchet & Murthy, 2019; Mohajerin Esfahani & Kuhn, 2018). The latter has demonstrated notable advantages in terms of efficiency and simplicity, in addition to being widely adopted in computer vision and graphics downstream applications (Rubner et al., 2000; Pele & Werman, 2008; Solomon et al., 2015; 2014), as it takes into account the geometry of the sample space contrarily to other metrics.

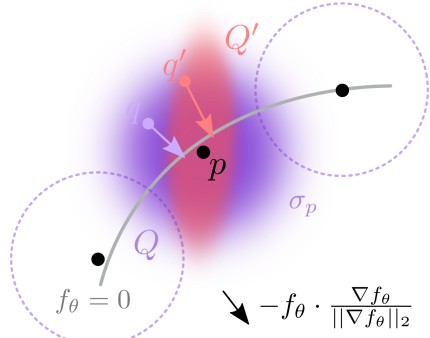

In order to learn a neural SDF from a sparse noisy point cloud withint a DRO framework we proceed in this work as follows. We first present a tractable implementation for this problem (SDF WDRO) benefiting from the dual reformulation (Blanchet & Murthy, 2019) of the DRO problem with Wasserstein distribution metric (Mohajerin Esfahani & Kuhn, 2018; Blanchet & Murthy, 2019; Sinha et al., 2017; Bui et al., 2022). We build on NP (Ma et al., 2021), but instead of using their predefined empirical spatial query distribution (sampling normally around each of the input points) we rely on queries from the worst-case distribution in the Wasserstein ball around the empirical distribution. While this leads to reduced overfitting and more robust reconstructions thanks to harnessing more informative samples midst training instead of overfitting on easy ones, this improvement comes at the cost of additional training time compared to the NP baseline as shown in Figure 7. Furthermore, by interpreting the Wasserstein distance computation as a mass transportation problem, recent advances in Optimal Transport show that it is possi-

Figure 1: We learn a neural SDF $f_\theta$ from a point cloud (black dots) by minimizing the error between projection of spatial queries $\{q\}$ on the level set of the field (gray curve) and their nearest input point $p$. Instead of learning with a standard predefined distribution of queries $Q$, we optimize for the worst-case query distribution $Q'$ within a ball of distributions around $Q$.

ble to obtain theoretically grounded approximations by regularizing the original mass transportation problem with relative entropy penalty on the transport plan (*e.g.* Cuturi (2013)). The resulting distance is referred to as Sinkhorn distance. Thus, we show subsequently that substituting the Wasserstein distance with the Sinkhorn one in our SDF DRO problem results in a computationally efficient dual formulation that significantly improves the convergence time of our first baseline SDF WDRO. The training Algorithm of the resulting SDF SDRO is outlined in 1.

Through extensive quantitative and qualitative evaluation under several real and synthetic benchmarks for object, non rigid and scene level shape reconstruction, our results show that our final method (SDF SDRO) outperforms SDF WDRO, baseline NP, as well as the most relevant competition, notably the current state-of-the-art in learning SDFs from sparse point cloud unsupervisedly NTPS (Chen et al., 2023a), NAP (Ouasfi & Boukhayma, 2024a). Our ablation studies the utility of distributional robustness in the context of unsupervised neural reconstruction from sparse input using pointwise adversaries.

**Summary of intuition and contribution** Our key idea is to construct a distribution of the most challenging query samples around the shape in terms of the loss function by "perturbing" the initial distribution of query points. The cost of this perturbation is controled globally through an optimal transport distance. Minimising the expected loss over this distribution flattens the landscape of the loss spatially ensuring that the implicit model behaves consistently in the 3D space. Not only does this act as a regularization but it additionally refines the implicit representation by providing informative samples throughout the training process.

## 2 RELATED WORK

**Reconstruction from Point Clouds** Classical approaches include combinatorical methods where shape is defined based on the input point cloud through space partitioning, using *e.g.* alpha shapes (Bernardini et al., 1999), Voronoi diagrams (Amenta et al., 2001), or triangulation (Cazals & Giesen, 2006; Liu et al., 2020; Rakotosaona et al., 2021). Alternatively, the input samples can define an implicit function, with its zero level set representing the target shape. This is achieved through global smoothing priors (Williams et al., 2022; Lin et al., 2022; Williams et al., 2021; Ouasfi & Boukhayma, 2024c), such as radial basis functions (Carr et al., 2001) and Gaussian kernel fitting (Schölkopf et al., 2004), or local smoothing priors like moving least squares (Mercier et al., 2022; Guennebaud & Gross, 2007; Kolluri, 2008; Liu et al., 2021). Another approach involves solving a boundary-conditioned Poisson equation (Kazhdan & Hoppe, 2013). Recent literature suggests parameterizing these implicit functions with deep neural networks and learning their parameters through gradient descent, either in a supervised (*e.g.* (Boulch & Marlet, 2022; Williams et al., 2022; Huang et al., 2023b; Peng et al., 2020; Chibane & Pons-Moll, 2020; Lionar et al., 2021; Ouasfi & Boukhayma, 2024b; Peng et al.,

2021)) or unsupervised manner. **Unsupervised Implicit Neural Reconstruction** A neural network is typically fitted to the a single point cloud without additional information in this setting. Improvements can be achieved through regularizations, such as the spatial gradient constraint based on the Eikonal equation introduced by Gropp et al. Gropp et al. (2020), a spatial Laplacian constraint as described in Ben-Shabat et al. (2022), and Lipschitz regularization on the network (Liu et al., 2022). Periodic activations were introduced in Sitzmann et al. (2020). Lipman (2021) learns a function that converges to occupancy, while its log transform converges to a distance function. Atzmon & Lipman (2020a) learn an SDF from unsigned distances, further supervising the spatial gradient of the function with normals (Atzmon & Lipman, 2020b). Ma et al. (2021) express the nearest point on the surface as a function of the neural signed distance and its gradient. They also utilize self-supervised local priors to handle very sparse inputs in Ma et al. (2022a) and enhance generalization in Ma et al. (2022b). Peng et al. (2021) proposed a differentiable Poisson solving layer that efficiently converts predicted normals into an indicator function grid. Koneputugodage et al. (2023) guides the implicit field learning with an Octree based labelling. Boulch et al. (2021) learns occupancy fields considering that needle end points close to the surface lie statistically on opposite sides of the surface. In Williams et al. (2021), infinitely wide shallow MLPs are learned as random feature kernels using points and their normals. Chen et al. (2023a) learns a surface parametrization leveraged to provide additional coarse surface supervision to the shape network. However, most of the aforementioned methods still encounter difficulties in learning suitable reconstructions when dealing with sparse and noisy input, primarily due to lack of adequate supervision. Ouasfi & Boukhayma (2024d) learn an occupancy function by sampling from it's uncertainty field and stabilise the optimization process by biasing the occupancy function towards minimal entropy fields. Ouasfi & Boukhayma (2024a) augment the training with adversarial samples around the input point cloud. Differently from this literature, **we explore here a new paradigm for learning unsupervised neural SDFs for the first time, namely through tractable reformulations of DRO**.

## 3 METHOD

Let $\Xi$ be a subset of $\mathbb{R}^3$. We denote the set of measures and the set of probabilities measures on $\Xi$ by $\mathcal{M}(\Xi)$, and $\mathcal{P}(\Xi)$ respectively. Given a noisy, sparse unoriented point cloud $\mathbf{P} \subset \Xi^{N_p}$, our objective is to obtain a corresponding watertight 3D shape reconstruction, *i.e.* the shape surface $\mathcal{S}$ that best explains the observation $\mathbf{P}$. In order to achieve this goal, we learn a shape function $f$ parameterised with an MLP $f_\theta$. The function represents the implicit signed distance field relative to the target shape $\mathcal{S}$. The inferred shape $\hat{\mathcal{S}}$ can be obtained as the zero level set of the SDF (signed distance function) $f_\theta$ at convergence: $\hat{\mathcal{S}} = \{q \in \mathbb{R}^3 \mid f_\theta(q) = 0\}$. Practically, an explicit triangle mesh for $\hat{\mathcal{S}}$ can be obtained through the Marching Cubes algorithm (Lorensen & Cline, 1987), while querying neural network $f_\theta$.

### 3.1 BACKGROUND: LEARNING AN SDF BY PULLING QUERIES ONTO THE SURFACE.

Neural Pull (NP) (Ma et al., 2021) approximates a signed distance function by pulling query points to their their nearest input point cloud sample using the gradient of the SDF network. The normalized gradient is multiplied by the negated signed distance predicted by the network in order to pull both inside and outside queries to the surface. Query points $q \in \mathfrak{Q}$ are sampled around the input point cloud $\mathbf{P}$, specifically from normal distributions centered at input samples $\{p\}$, with locally defined standard deviations $\{\sigma_p\}$:

$$\mathfrak{Q} := \bigcup_{p \in \mathbf{P}} \{q \sim \mathcal{N}(p, \sigma_p \mathbf{I}_3)\}, \tag{1}$$

where $\sigma_p$ is defined as the maximum euclidean distance to the $K$ nearest points to $p$ in $\mathbf{P}$. For each query $q$, the nearest point $p$ in $\mathbf{P}$ is computed subsequently, and the following objective is optimized in Ma et al. (2021) yielding a neural SDF $f_\theta$ whose zero level set concurs with the samples in $\mathbf{P}$:

$$\mathcal{L}(\theta, q) = ||q - f_\theta(q) \cdot \frac{\nabla f_\theta(q)}{||\nabla f_\theta(q)||_2} - p||_2^2, \quad \text{where} \quad p = \underset{t \in \mathbf{P}}{\arg \min} ||t - q||_2. \tag{2}$$

The SDF network is trained with empirical risk minimization (ERM) by minimizing the expected loss under the empirical distribution $Q = \sum_{q \in \mathfrak{Q}} \delta_q$ over the set Q where $\delta_q$ is the dirac distribution or the unit mass on $q$.

## 3.2 Neural SDF DRO

Inspired by Ouasfi & Boukhayma (2024a), we focus on how to distribute the SDF approximation errors uniformly throughout the shape as these erros tends to concentrate low-density and noisy areas without regularization. We consider the DRO problem introduced by NAP with Wasserstein uncertainty sets. This restrains the set of worst-case distributions using the Wasserstein distance (Eq. 3). We optimize the parameters of the SDF network $\theta$ under the worst-case expected loss among a ball of distributions $Q'$ in this uncertainty set (Gao & Kleywegt, 2023; Blanchet & Murthy, 2019),:

$$\inf_{\theta} \sup_{Q':\mathcal{W}_c(Q',Q)<\epsilon} \mathbb{E}_{q'\sim Q'} \mathcal{L}(\theta, q'), \text{ where } \mathcal{W}_c(Q',Q) := \inf_{\gamma\in\Gamma(Q',Q)} \int cd\gamma. \tag{3}$$

Here, $\epsilon > 0$ and $\mathcal{W}_c$ denotes the optimal transport (OT) or a Wasserstein distance for a cost function $c$, defined as the infimum over the set $\Gamma(Q',Q)$ of couplings whose marginals are $Q'$ and $Q$. We refer the reader to the body of work in *e.g.* Gao & Kleywegt (2023); Blanchet & Murthy (2019) for more background.

**Neural SDF Wasserstein DRO (WDRO)** A tractable reformulation of the optimization problem defined in Equation 3 is made possible thanks to the following duality result (Blanchet & Murthy, 2019). For upper semi-continuous loss functions and non-negative lower semi-continuous costs satisfying $c(z, z') = 0$ iff $z = z'$, the optimization problem (3) is equivalent to:

$$\inf_{\theta,\lambda\geq 0} \{\lambda\epsilon + \mathcal{L}_{\text{WDRO}}(\theta, Q)\}, \text{ where } \mathcal{L}_{\text{WDRO}}(\theta, Q) = \mathbb{E}_{q\sim Q}\left[\sup_{q'}\{\mathcal{L}(\theta, q') - \lambda c(q', q)\}\right]. \tag{4}$$

As shown in Bui et al. (2022), solving the optimization above with a fixed dual variable $\lambda$ yields inferior results to the case where $\lambda$ is updated. In fact, optimizing $\lambda$ allows to capture global information when solving the outer minimization, whilst only local information (local worst-case spatial queries) is considered when minimizing $\mathcal{L}_{\text{WDRO}}$ solely. Following Bui et al. (2022), the optimization in Equation 4 can be carried as follows: Given the current model parameters $\theta$ and the dual variable $\lambda$, the worst-case spatial query $q'$ corresponding to a query $q$ drawn from the empirical distribution $Q$ can be obtained through a perturbation of $q$ followed by a few steps of iterative gradient ascent over $\mathcal{L}(\theta, q') - \lambda c(q', q)$. Subsequently, inspired by the Danskin's theorem we can update $\lambda$ accordingly $\lambda \leftarrow \lambda - \eta_\lambda \left(\epsilon - \frac{1}{N_b}\sum_{i=1}^{N_b} c(q'_i, q_i)\right)$, where $N_b$ represents the query batch size, and $\eta_\lambda > 0$ symbolizes a learning rate. The current batch loss $\mathcal{L}_{\text{WDRO}}$ can then be backpropagated. We provide an Algorithm in supplemental material recapitulating this training (2).

To sample from the worst case distribution around the shape, WDRO (Equation 4) relies on a soft-ball projection controlled by the parameter $\lambda$ that is adjusted throughout the training. The $\lambda$ update rule ensures that it grows when the worst-case sample distance from the initial queries exceeds the Wasserstein ball radius $\epsilon$. In contrast, NAP consists of a hard-ball projection with locally adaptive radius.

While WDRO provides promising results, it suffers from rather slow convergence, as shown in Figure 7. Furthermore, because our nominal distribution $Q$ is finitely supported, the worst-case distribution generated with WDRO is proven to be a discrete distribution (Gao & Kleywegt, 2023), even while the underlying actual distribution is continuous. As pointed out in Wang et al. (2021), concerns emerge around whether WDRO hedges the right family of distributions and generates solutions that are too conservative. In the next section, we show how these limitations can be addressed by taking inspiration from recent advances in Optimal Transport (OT).

**Neural SDF Wasserstein DRO with entropic regularization (SDRO)** One key technical aspect underpinning the recent achievements of Optimal Transport (OT) in various applications lies in the use of regularization, particularly entropic regularization. This approach has paved the way for efficient computational methodologies (see *e.g.* Cuturi (2013)) to obtain theoretically-grounded approximations of Wasserstein distances. Building upon these advancements, recent work (Azizian et al., 2023; Wang et al., 2021) extend the framework of Wasserstein Distributionally Robust Optimization with entropic regularization by substituting the Wasserstein distance in Equation 3 with the Sinkhorn distance (Wang et al., 2021).

For $P, Q \in \mathcal{P}(\Xi)$, and two reference measures $\mu, \nu \in \mathcal{M}(\Xi)$ such that $P$ and $Q$ are absolutely continuous to $\mu$ and $\nu$ respectively, the Sinkhorn distance is defined as:

$$\mathcal{W}_\rho(P,Q) = \inf_{\gamma\in\Gamma(P,Q)}\{\mathbb{E}_{(x,y)\sim\gamma}[c(x,y)] + \rho H(\gamma \mid \mu\otimes\nu)\}, \tag{5}$$

where $\rho \geq 0$ is a regularization parameter. $H(\gamma \mid \mu \otimes \nu)$ denotes the relative entropy of $\gamma$ with respect to the product measure $\mu \otimes \nu$ :

$$H(\gamma \mid \mu \otimes \nu) = \mathbb{E}_{(x,y) \sim \gamma} \left[ \log \left( \frac{\mathrm{d}\gamma(x,y)}{\mathrm{d}\mu(x)\mathrm{d}\nu(y)} \right) \right], \tag{6}$$

where $\frac{\mathrm{d}\gamma(x,y)}{\mathrm{d}\mu(x)\mathrm{d}\nu(y)}$ stands for the density ratio of $\gamma$ with respect to $\mu \otimes \nu$ evaluated at $(x, y)$.

Compared to the Wasserstein distance, Sinkhorn distance regularizes the original mass transportation problem with relative entropy penalty on the transport plan. The choice of the reference measures $\mu$ and $\nu$ acts as a prior on the DRO problem. Following Wang et al. (2021), we fix $\mu$ as our empirical distribution $Q$ and $\nu$ as the Lebesgue measure. Consequently, optimization problem in Equation 3 with the Sinkhorn distance admits the following dual form:

$$\inf_{\theta, \lambda \geq 0} \left\{ \lambda \bar{\epsilon} + \lambda \rho \mathbb{E}_{q \sim Q} \left[ \log \mathbb{E}_{q' \sim \mathbb{Q}_{q,\rho}} \left[ e^{\mathcal{L}(\theta, q')/(\lambda\rho)} \right] \right] \right\}, \tag{7}$$

where $\bar{\epsilon}$ is a constant that depends on $\rho$ and $\epsilon$ (Wang et al. (2021)). Additionally, distribution $\mathbb{Q}_{q,\rho}$ is defined through:

$$\mathrm{d}\mathbb{Q}_{x,\rho}(z) := \frac{e^{-c(x,z)/\rho}}{\mathbb{E}_{u \sim \nu} \left[ e^{-c(x,u)/\rho} \right]} \mathrm{d}\nu(z). \tag{8}$$

As discussed in Wang et al. (2021), optimizing $\lambda$ within problem 7 leads to instability. Hence, for a given fixed $\lambda > 0$, optimization 7 can be carried practically by sampling a set of $N_s$ samples $q' \sim \mathbb{Q}_{q,\rho}$ for each query $q$, then backpropagating the following distributionaly robust loss:

$$\mathcal{L}_{\mathrm{SDRO}}(\theta, Q) = \lambda \rho \mathbb{E}_{q \sim Q} \left[ \log \mathbb{E}_{q' \sim \mathbb{Q}_{q,\rho}} \left[ e^{\mathcal{L}(\theta, q')/(\lambda\rho)} \right] \right]. \tag{9}$$

Algorithm 1 summarizes the training of our SDRO based method.

### 3.3 Training Objective

Similar to Ouasfi & Boukhayma (2024a) we train using the strategy of Liebel & Körner (2018) which combines the original objective and the distributionally robust one:

$$\mathfrak{L}(\theta, q) = \frac{1}{2\lambda_1} \mathcal{L}(\theta, q) + \frac{1}{2\lambda_2} \mathcal{L}_{\mathrm{DRO}}(\theta, q) + \ln(1 + \lambda_1) + \ln(1 + \lambda_2), \tag{10}$$

where $\lambda_1$ and $\lambda_2$ are learnable weights and $\mathcal{L}_{\mathrm{DRO}}$ is either $\mathcal{L}_{\mathrm{SDRO}}$ or $\mathcal{L}_{\mathrm{WDRO}}$. Sur training procedure is shown in Algorithms 2 and 1.

---

**Algorithm 1** The training procedure of our method with SDRO.

---

**Input:** Point cloud $\mathbf{P}$, learning rate $\alpha$, number of iterations $N_{\mathrm{it}}$, batch size $N_b$.
    SDRO hyperparameters: $\rho, \lambda, N_s$.
**Output:** Optimal parameters $\theta^*$.
    Compute local st. devs. $\{\sigma_p\}$ ($\sigma_p = \max_{t \in K_{\mathrm{nn}}(p, \mathbf{P})} ||t - p||_2$).
    $\mathfrak{Q} \leftarrow \mathrm{sample}(\mathbf{P}, \{\sigma_p\})$ (Equ. 1)
    Compute nearest points in $\mathbf{P}$ for all samples in $\in \mathfrak{Q}$.
    Initialize $\lambda_1 = \lambda_2 = 1$.
    **for** $N_{\mathrm{it}}$ times **do**
        Sample $N_b$ query points $\{q, q \sim Q\}$.
        For each $q$, sample $N_s$ points $\{q', q' \sim \mathbb{Q}_{q,\rho}\}$. (Equ.8)
        Compute SDRO losses $\{\mathcal{L}_{\mathrm{SDRO}}(\theta, q)\}$ (Equ. 9)
        Compute combined losses $\{\mathfrak{L}(\theta, q)\}$ (Equ. 10)
        $(\theta, \lambda_1, \lambda_2) \leftarrow (\theta, \lambda_1, \lambda_2) - \alpha \nabla_{\theta, \lambda_1, \lambda_2} \Sigma_q \mathfrak{L}(\theta, q)$
    **end for**

---

## 4 Results

We evaluate our method using standard reconstruction benchmarks. Following previous work, we compute the accuracy of the 3D meshes extracted from our MLPs at convergence. We compare to state of the art methods dedicated to sparse unsupervised reconstruction NP(Ma et al., 2021),NAP (Ouasfi & Boukhayma, 2024a), SparseOcc (Ouasfi & Boukhayma, 2024d) and NTPS Chen et al. (2023a). We additionally compare to SAP (Peng et al., 2021), DIGS (Ben-Shabat et al., 2022),

NDrop (Boulch et al., 2021), NSpline (Williams et al., 2021) and methods combining explicit and implicit representations such as OG-INR (Koneputugodage et al., 2023) and GP (GridPull) (Chen et al., 2023b). We further compare to supervised methods including state of the art feed-forward generalizable methods POCO (Boulch & Marlet, 2022), CONet (Peng et al., 2020) and NKSR (Huang et al., 2023a), and the prior-based optimization method dedicated to sparse inputs On-Surf (Ma et al., 2022a). Following NAP, we experimented with point clouds of size $N_p = 1024$.

### 4.1 METRICS

We use standard metrics for the 3D reconstruction task. We compute the L1 **Chamfer Distance** $CD_1$ ($\times 10^2$), L2 **Chamfer Distance** $CD_2$ ($\times 10^2$), the euclidean distance based **F-Score (FS)** and **Normal Consistency** (NC) between our extracted mesh and the ground-truth. The corresponding mathematical expressions are provided in the the supplementary material.

### 4.2 DATASETS AND INPUT DEFINITIONS

**ShapeNet** (Chang et al., 2015) includes a wide range of synthetic 3D objects spanning 13 different categories. Following NAP, we show results on classes Tables, Chairs and Lamps using the train/test splits defined in Williams et al. (2021). We generate noisy input point clouds by sampling 1024 points from the meshes while adding Gaussian noise of standard deviation 0.005 (*e.g.* Boulch & Marlet (2022); Peng et al. (2020); Ouasfi & Boukhayma (2024a)). **Faust** (Bogo et al., 2014) consists of real scans of 10 human body identities in 10 different poses. We sample sets of 1024 points from the scans as inputs. **3D Scene** (Zhou & Koltun, 2013) consists of large scale complex real world scenes obtained with a handheld commodity range sensor. We follow Chen et al. (2023a); Jiang et al. (2020); Ma et al. (2021); Ouasfi & Boukhayma (2024a) to sample sparse point clouds with a density of 100 points per m$^3$ and report results. We show results for scenes Burghers, Copyroom, Lounge, Stonewall and Totempole. **Surface Reconstruction Benchmark (SRB)** (Williams et al., 2019) is made of five object scans with complex topology, high level of detail, missing data and varying feature scales. We sample 1024 points from the scans for the sparse input experiment, and we experiment using the dense inputs as well. **SemanticPOSS** Pan et al. (2020) consists of 6 sequences of road scene LiDAR data. Each scan covers a range of 51.2m ahead of the LiDAR, 25.6m to each side, and 6.4m in height. We show qualitative examples from each sequence. We further test our method on few challenging scenes from **BlendedMVS** (Yao et al., 2020) and on large-scale scenes from **Tanks Temples dataset** (Knapitsch et al., 2017) with sparse views.

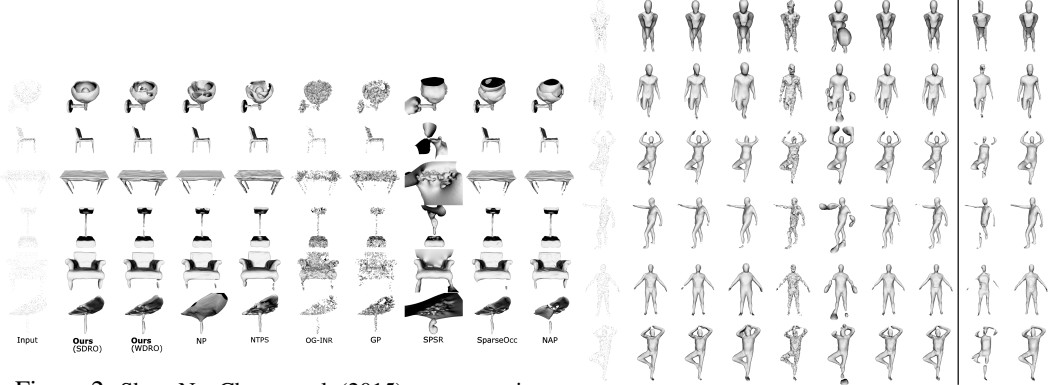

Figure 2: ShapeNet Chang et al. (2015) reconstructions.

Figure 3: Faust Bogo et al. (2014) reconstructions. **CONet and POCO use data priors**.

### 4.3 IMPLEMENTATION DETAILS

Our MLP model, ($f_\theta$), follows the architecture specified in Neural Pull (NP) (Ma et al., 2021). We optimize the model using the Adam optimizer with a batch size of $N_b = 5000$. Consistent with NP, we set $K = 51$ to compute local standard deviations $\sigma_p$. Training is conducted on a single NVIDIA RTX A6000 GPU. To ensure fairness and practicality in our comparative evaluation, we identify the optimal evaluation epoch for each method based on the Chamfer distance between the reconstructed and input point clouds, selecting the best-performing epoch under this metric. Using this validation criterion, we conduct a hyperparameter search on the SRB benchmark to determine

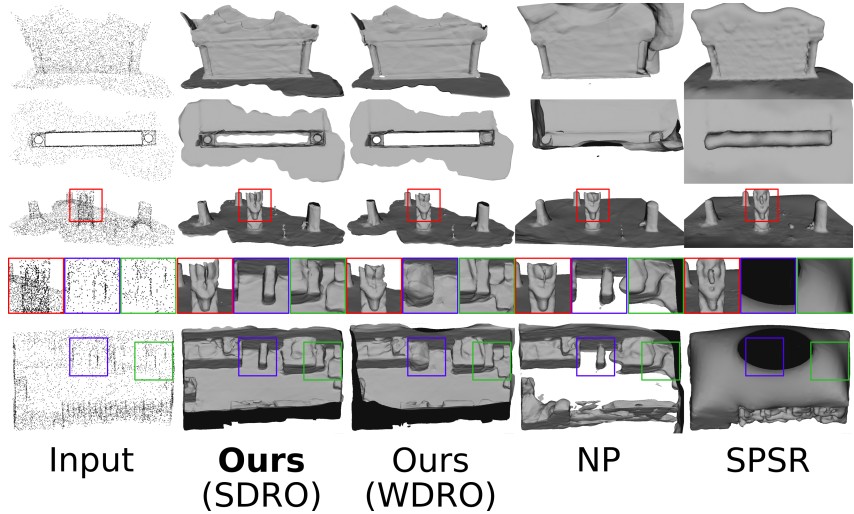

Figure 4: 3D Scene (Zhou & Koltun, 2013) reconstructions from sparse unoriented point clouds.

the optimal parameters for our methods. For the Wasserstein Robust DRO (WRDO) approach, we perform $N_{it}^{wdro} = 2$ gradient ascent steps in the inner loop with a learning rate of $\alpha_{wdro} = 10^{-3}$. The dual variable is initialized to $\lambda = 80$, and the Wasserstein ball radius is fixed at $\epsilon = 10^{-4}$. For the Standard DRO (SDRO) approach, we use $N_s = 5$ samples for each query point $q \sim Q$, with $\lambda = 20$ in our experiments. We define the transport cost as $c(\cdot, \cdot) = \frac{1}{2}||\cdot - \cdot||^2$, which implies that sampling from $\mathbb{Q}_{q,\rho}$ corresponds to sampling from a Gaussian distribution $\mathcal{N}(q, \rho\mathbf{I}_3)$.

### 4.4 OBJECT LEVEL RECONSTRUCTION

We perform reconstruction of ShapeNet (Chang et al., 2015) objects from sparse and noisy point clouds. Table 1 and Figure 2 show respectively a numerical and qualitative comparison to the competition. While our WDRO-based method demonstrates superior performance compared to competitors in terms of reconstruction accuracy, as assessed by $CD_1$ and $CD_2$, incorporating the SDRO loss further enhances performance across all metrics. This is evidenced by the visually superior quality of our reconstructions, which exhibit improved fidelity in capturing fine structures and details. Despite achieving generally satisfactory coarse reconstructions, the thin plate spline smoothing prior utilized by NTPS appears to limit its expressiveness. Additionally, we observed that OG-INR struggles to converge to satisfactory results under sparse and noisy conditions, despite its effective guidance from Octree-based sign fields in denser settings.

|  | CD1 | CD2 | NC | FS |
|---|---|---|---|---|
| SPSR | 2.34 | 0.224 | 0.74 | 0.50 |
| OG-INR | 1.36 | 0.051 | 0.55 | 0.55 |
| NP | 1.16 | 0.074 | 0.84 | 0.75 |
| GP | 1.07 | 0.032 | 0.70 | 0.74 |
| NTPS | 1.11 | 0.067 | 0.88 | 0.74 |
| NAP | 0.76 | 0.020 | 0.87 | 0.83 |
| SparseOcc | 0.76 | 0.020 | 0.88 | 0.83 |
| Ours (WDRO) | 0.77 | 0.015 | 0.87 | 0.83 |
| **Ours** (SDRO) | 0.63 | 0.012 | 0.90 | 0.86 |

Table 1: ShapeNet (Chang et al., 2015) reconstructions from sparse noisy unoriented point clouds.

|  | CD1 | CD2 | NC | FS |
|---|---|---|---|---|
| POCO | 0.308 | 0.002 | 0.934 | 0.981 |
| CONet | 1.260 | 0.048 | 0.829 | 0.599 |
| On-Surf | 0.584 | 0.012 | 0.936 | 0.915 |
| NKSR | 0.274 | 0.002 | 0.945 | 0.981 |
| SPSR | 0.751 | 0.028 | 0.871 | 0.839 |
| GP | 0.495 | 0.005 | 0.887 | 0.945 |
| NTPS | 0.737 | 0.015 | 0.943 | 0.844 |
| NAP | 0.220 | 0.001 | 0.956 | 0.981 |
| SparseOcc | 0.260 | 0.002 | 0.952 | 0.974 |
| Ours (WDRO) | 0.255 | 0.002 | 0.953 | 0.977 |
| **Ours** (SDRO) | 0.251 | 0.002 | 0.955 | 0.979 |

Table 2: Faust (Bogo et al., 2014) reconstructions from sparse noisy unoriented point clouds. **POCO, CONet, On-Surf and NKSR use data priors.**

### 4.5 REAL ARTICULATED SHAPE RECONSTRUCTION

We conduct the reconstruction of Faust (Bogo et al., 2014) human shapes using sparse and noisy point clouds. Competing approaches are compared both quantitatively and qualitatively in Table 2 and Figure 3. All evaluation metrics show that our distributionally robust training procedures work better. Using SDRO leads to a marginally better performance and noticeably faster convergence as compared to training with the WDRO loss. Our reconstructions are visually far better, especially when it comes to catching details at the extremities of the body. These extremities present difficulties because of sparse input point cloud data, which leads to confusing form prediction, much as the fine structures shown in the ShapeNet experiment. Interestingly, our method is outperformed by NAP in this setting and performs on par with SparseOcc. This expected as these methods work well under small levels of

| | Burghers | | | Copyroom | | | Lounge | | | Stonewall | | | Totemple | | | Mean | | |
|---|---|---|---|---|---|---|---|---|---|---|---|---|---|---|---|---|---|---|
| | CD1 | CD2 | NC | CD1 | CD2 | NC | CD1 | CD2 | NC | CD1 | CD2 | NC | CD1 | CD2 | NC | CD1 | CD2 | NC |
| SPSR | 0.178 | 0.2050 | 0.874 | 0.225 | 0.2860 | 0.861 | 0.280 | 0.3650 | 0.869 | 0.300 | 0.4800 | 0.866 | 0.588 | 1.6730 | 0.879 | 0.314 | 0.6024 | 0.870 |
| NDrop | 0.200 | 0.1140 | 0.825 | 0.168 | 0.0630 | 0.696 | 0.156 | 0.0500 | 0.663 | 0.150 | 0.0810 | 0.815 | 0.203 | 0.1390 | 0.844 | 0.175 | 0.0894 | 0.769 |
| NP | 0.064 | 0.0080 | 0.898 | 0.049 | 0.0050 | 0.828 | 0.133 | 0.0380 | 0.847 | 0.060 | 0.0050 | 0.910 | 0.178 | 0.0240 | 0.908 | 0.097 | 0.0160 | 0.878 |
| SAP | 0.153 | 0.1010 | 0.807 | 0.053 | 0.0090 | 0.771 | 0.134 | 0.0330 | 0.813 | 0.070 | 0.0070 | 0.867 | 0.474 | 0.3820 | 0.725 | 0.151 | 0.1064 | 0.797 |
| NSpline | 0.135 | 0.1230 | 0.891 | 0.056 | 0.0230 | 0.855 | 0.063 | 0.0390 | 0.827 | 0.124 | 0.0910 | 0.897 | 0.378 | 0.7680 | 0.892 | 0.151 | 0.2088 | 0.872 |
| NTPS | 0.055 | 0.0050 | 0.909 | 0.045 | 0.0030 | 0.892 | 0.129 | 0.0220 | 0.872 | 0.054 | 0.0040 | 0.939 | 0.103 | 0.0170 | 0.935 | 0.077 | 0.0102 | 0.897 |
| NAP | 0.051 | 0.006 | 0.881 | 0.037 | 0.002 | 0.833 | 0.044 | 0.011 | 0.862 | 0.035 | 0.003 | 0.912 | 0.042 | 0.002 | 0.925 | 0.041 | 0.004 | 0.881 |
| SparseOcc | 0.022 | 0.001 | 0.871 | 0.041 | 0.012 | 0.812 | 0.021 | 0.001 | 0.870 | 0.028 | 0.003 | 0.931 | 0.026 | 0.001 | 0.936 | 0.027 | 0.003 | 0.886 |
| **Ours** (WDRO) | 0.014 | 0.0006 | 0.871 | 0.028 | 0.0036 | 0.820 | 0.038 | 0.0051 | 0.803 | 0.019 | 0.0005 | 0.930 | 0.009 | 0.0003 | 0.936 | 0.022 | 0.0020 | 0.872 |
| **Ours** (SDRO) | 0.015 | 0.0006 | 0.873 | 0.021 | 0.0017 | 0.823 | 0.027 | 0.0032 | 0.842 | 0.021 | 0.0006 | 0.932 | 0.020 | 0.0005 | 0.934 | 0.020 | 0.0013 | 0.881 |

Table 3: 3D Scene (Zhou & Koltun, 2013) reconstructions from sparse point clouds.

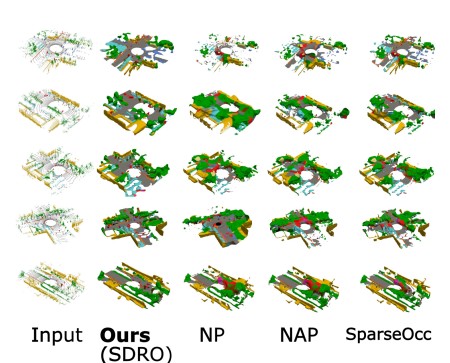

Figure 5: SemanticPOSS Pan et al. (2020) reconstruction from road scene LiDAR data.

Figure 6: Qualitative comparisons on BlendedMVS (Yao et al., 2020) and Tanks Temples dataset (Knapitsch et al., 2017).

noise while ours is dedicated to high levels of noise. On the other hand, NTPS reconstructions are typically coarser and less detailed. It should be noted that not every ShapeNet-trained Generalizable method (seen in the table's upper section) performs well in this particular experiment.

### 4.6 REAL SCENE LEVEL RECONSTRUCTION

We present reconstruction results on the 3D Scene (Zhou & Koltun, 2013) data from sparse point clouds following Chen et al. (2023a). Results for the state-of-the-art NTPS technique, NP, SAP, NDrop, and NSpline were compiled from NTPS. Results for NAP and SparseOcc are reported from their respective papers and summarized in Table 3. We outperform the competition in this setting because of our loss that can hedge against high levels of noise in contrast to NAP. The qualitative comparisons to our baseline NP and SPSR are displayed in Figure 4. Areas where our technique exhibits particularly excellent details and fidelity in the reconstruction are shown by colored boxes.

Additionally, we conduct qualitative comparisons on BlendedMVS (Yao et al., 2020) and large-scale scenes from the Tanks  Temples dataset (Knapitsch et al., 2017) using sparse views. VGGSfM (Wang et al., 2024), a recent state-of-the-art fully differentiable structure-from-motion pipeline, is used for this experiment. Although VGGSfM effectively generates point clouds by triangulating 2D point trajectories and learned camera poses, the sparse input images result in sparse and noisy point clouds, making SDF-based reconstruction challenging. To illustrate the strength of our method, we compare 3 examples from each dataset against SparseOcc and NAP in Fig. , demonstrating sharper details, especially on large-scale scenes from Tanks  Temples, where other methods struggle due to noise in VGGSfM's point clouds.

To further assess the robustness of our approach, we show reconstructions under the SemanticPOS dataset Pan et al. (2020) and provide qualitative comparisons with SparseOcc, NAP , and NP (Ma et al., 2021) in Figure 5. The visualizations use color-coded semantic segmentations from the dataset, which are not used during training. Our results demonstrate a clear improvement in reconstruction quality, attributable to our DRO formulation. Notably, objects such as cars, trees, and pedestrians are reconstructed with greater detail and accuracy, while the baseline methods tend to merge these object classes into indistinct blobs. Our method (SDRO) also excels in preserving the overall scene layout. Notably, while SparseOcc and NAP perform well under low noise, their performance degrades significantly under high noise levels. More qualitative results are available in the supplementary material.

Figure 7: Performance over training time on Shapenet (Chang et al., 2015) class Tables.

| | $\sigma = 0.0$ | | $\sigma = 0.005$ | | $\sigma = 0.025$ | |
|---|---|---|---|---|---|---|
| | CD1 | NC | CD1 | NC | CD1 | NC |
| NP (baseline) | 0.73 | 0.906 | 1.07 | 0.847 | 2.45 | 0.668 |
| NAP | 0.63 | 0.926 | 0.75 | 0.86 | 2.21 | 0.67 |
| SparseOcc | 0.56 | 0.931 | 0.77 | 0.89 | 2.16 | 0.68 |
| Ours(SDRO) | 0.43 | 0.945 | 0.65 | 0.91 | 1.54 | 0.702 |

Table 4: Ablation of our method under varying levels of noise on Shapenet (Chang et al., 2015) class Tables.

## 5    ABLATION STUDIES

**Noise ablation** To isolate the impact of input sparsity and input noise (displacement relative to the surface) on our method's performance compared to the NP baseline, we present results with varying noise levels in Table 4. These findings consistently demonstrate our method's improvement over the baseline across different levels of noise. This suggests that our distributionally robust training strategy effectively mitigates noise in the labels arising from both input displacement from the surface and input sparsity. We note that with high levels of noise our approach outperforms NAP and SparseOcc.

**Training time** To assess the computational efficiency of our method, we present, in Figure 7, the performance improvement over training time for our proposed DRO approaches as well as the NP baseline. This graph illustrates the performance achieved by training for specific durations. Specifically, we observe that WDRO reaches the NP baseline performance with a delay of 3 minutes and requires a total of 10 minutes to achieve its peak performance. In contrast, SDRO shows improvement over the NP baseline after training for only 2 minutes and reaches its best performance in less than 6 minutes, matching the convergence time of the baseline while significantly improving on both the baseline and our WDRO approach performance. This highlights the computational benefits of relying on the Sinkhorn distance instead of Wasserstein distance in defining the uncertainty sets of our distributional robust optimization problem (3). Additional ablation studies are provided in the supplemental material.

## 6    LIMITATIONS

In some specific settings it can be hard to set the hyperparameters of our method. Increasing the dual variable $\lambda$ that controls the perturbation cost can result in under-performance. While this limitation is shared with NAP (Ouasfi & Boukhayma, 2024a), it's not clear how to combine the adaptive local control on spatial adversaries provided by NAP (Ouasfi & Boukhayma, 2024a) with the global control provided by WDRO and SDRO. We plan to address this point in future work.

## 7    CONCLUSION

We have shown that regularizing implicit shape representation learning from sparse unoriented point clouds through distributionally robust optimization can lead to superior reconstructions. We believe these new findings can usher in a new body of work incorporating distributional robustness in learning various forms of neural implicits, which in turn can potentially have a larger impact beyond the specific scope of this paper.

## 8    POTENTIAL BROADER IMPACT

This paper presents work whose goal is to advance the fields of Machine Learning and 3D Computer Vision, specifically implicit neural shape representation learning. There are many potential societal consequences of our work, none of which we feel must be specifically highlighted here.

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

## ADDITIONAL RESULTS

To accompany numerical results in Table 5 in the main paper using the SRB (Williams et al., 2019) benchmark, we show here a qualitative comparison between our method and methods NP(Ma et al., 2021) (our beseline), OG-INR(Koneputugodage et al., 2023) and SPSR (Kazhdan & Hoppe, 2013). Notice that we recover better shapes overall.

## ADDITIONAL ABLATIVE ANALYSIS

| | Sparse | Dense |
|---|---|---|
| SPSR | 2.27 | 1.25 |
| DIGS (Ben-Shabat et al., 2022) | 0.68 | 0.19 |
| OG-INR (Koneputugodage et al., 2023) | 0.85 | 0.20 |
| NTPS (Chen et al., 2023a) | 0.73 | - |
| NP (Ma et al., 2021) | 0.58 | 0.23 |
| Ours (WDRO) | 0.51 | 0.20 |
| Ours (SDRO) | 0.48 | 0.21 |

Table 5: Ablation of point cloud density

**Varying the point cloud density** We use the SRB benchmark (Williams et al., 2019) to evaluate the performance of our method across various point cloud densities. Qualitative results in SRB are provided in Figure 8. Table 5 presents comparative results for both 1024-sized and dense input point

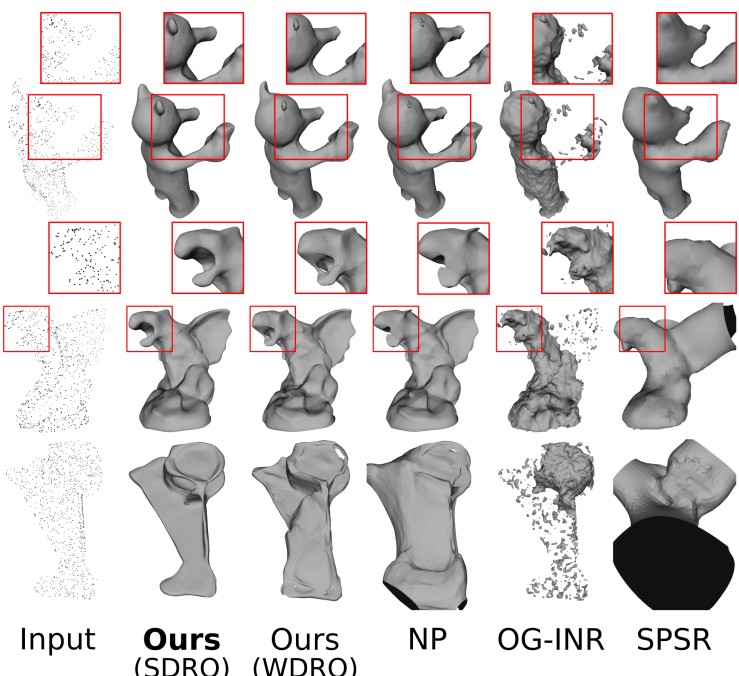

Figure 8: SRB (Williams et al., 2019) unsupervised reconstructions from sparse (1024 pts) unoriented point clouds without data priors.

clouds. We include results from the competition, specifically OG-INR, in the dense setting. Our distributionally robust training strategy outperforms competitors in the sparse case and performs comparably with the state-of-the-art in the dense case. Notably, we observe substantial improvement over our baseline (NP) for both sparse and dense inputs. These results underscore the practical utility and benefit of our contribution, even in dense settings. Interestingly, our ablation analysis reveals that for dense inputs, WDRO may exhibit slightly better performance compared to SDRO. This result is not surprising, given that WDRO is certified to effectively hedge against small perturbations (Sinha et al., 2017). Consequently, as the input becomes denser, the noise on the labels due to input sparsity diminishes, thereby favoring WDRO.

In order to determine the hyperparameters of our proposed approach (SDRO), We performed a hyperparameter search on the SRB (Williams et al., 2019) benchmark utilizing the chamfer distance between the reconstruction and the input point cloud as a validation metric. For the remaining datasets, we employed the same hyperparameters.

We carry out here an ablation study where we vary each one of the hyperparameters $\lambda$ and $\rho$ while fixing the remaining ones in order to better understand the behavior of our approach (SDRO) and its sensitivity to the choice of these hyperparameters.

**Regularization parameter** $\lambda$. This parameter controls how close the worst-case distribution $Q'$ is to the nominal distribution. As a result, Figure 10 illustrates how a very high value for this parameter minimizes the regularization impacts of SDRO by maintaining the worst-case samples around the nominal samples. Conversely, excessively low values lead to overly pessimistic estimations over-smoothing the results, despite greatly improving over the NP baseline.

**Regularization parameter** $\rho$. This parameter is responsible for the strength of the entropic regularization: it controls how the SDRO worst case distribution is concentrated around the support points of WDRO worst case distribution Wang et al. (2021). Consequently, it has to be defined such that it facilitates finding challenging distributions around the surface while maintaining a useful supervision signal. According to Figure 9, it is important to utilize a sufficiently high $\rho$ value in order to hedge against the right family of distributions. Contrastively, very high values can result in increased variance. Notice that $\rho_{avg}$ here corresponds to average $\sigma_p$ over the input points $\mathbf{P}$.

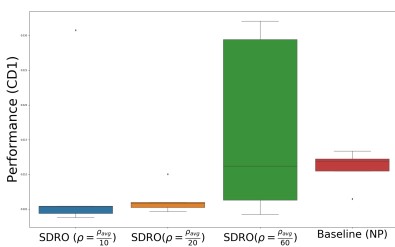 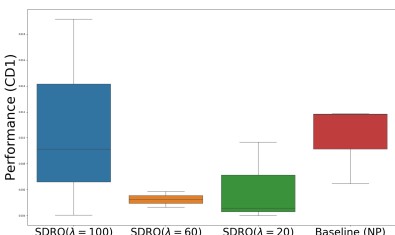

Figure 9: Ablation of the regularization parameter $\rho$.

Figure 10: Ablation of the regularization parameter $\lambda$.

## TRAINING ALGORITHM FOR WDRO

We provide in 2 the detailed training procedure for WDRO.

---

**Algorithm 2** The training procedure of our method with WDRO.

---

**Input:** Point cloud $\mathbf{P}$, learning rate $\alpha$, number of iterations $N_{\text{it}}$, batch size $N_b$.
  WDRO hyperparameters: $\epsilon$, $\sigma_0$, $\alpha_{wdro}$, $N_{it}^{wdro}$, $\eta_\lambda$.
**Output:** Optimal parameters $\theta^*$.
  Compute local st. devs. $\{\sigma_p\}$ $(\sigma_p = \max_{t \in K\text{nn}(p,\mathbf{P})} ||t - p||_2)$.
  $\mathfrak{Q} \leftarrow \text{sample}(\mathbf{P}, \{\sigma_p\})$. (Equ. 1)
  Compute nearest points in $\mathbf{P}$ for all samples in $\mathfrak{Q}$.
  Initialize $\lambda_1 = \lambda_2 = 1$.
  Initialize $\lambda$.
  **for** $N_{\text{it}}$ times **do**
    Sample $N_b$ query points $\{q, q \sim Q\}$.
    Initialize $N_b$ points $\{q'\}$, $(q' \sim \mathcal{N}(q, \sigma_0 \mathbf{I}_3))$.
    **for** $N_{\text{it}}^{wdro}$ times **do**
      $q' \leftarrow q' + \alpha_{wdro} \nabla_{q'} [\mathcal{L}(\theta, q') - \lambda c(q, q')]$
    **end for**
    $\lambda \leftarrow \lambda - \eta_\lambda \left( \epsilon - \frac{1}{N_b} \sum_{i=1}^{N_b} c(q'_i, q_i) \right)$
    Compute WDRO losses $\{\mathcal{L}_{\text{WDRO}}(\theta, q)\}$ (Equ. 4)
    Compute combined losses $\{\mathfrak{L}(\theta, q)\}$ (Equ. 10)
    $(\theta, \lambda_1, \lambda_2) \leftarrow (\theta, \lambda_1, \lambda_2) - \alpha \nabla_{\theta, \lambda_1, \lambda_2} \Sigma_q \mathfrak{L}(\theta, q)$
  **end for**

---

## EVALUATION METRICS

Following the definitions from Boulch & Marlet (2022) and Williams et al. (2019), we present here the formal definitions for the metrics that we use for evaluation in the main submission. We denote by $\mathcal{S}$ and $\hat{\mathcal{S}}$ the ground truth and predicted mesh respectively. We follow Chen et al. (2023a) to approximate all metrics with 100k samples from $\mathcal{S}$ and $\hat{\mathcal{S}}$ for ShapeNet and Faust and with 1M samples for 3Dscene. For SRB, we use 1M samples following Ben-Shabat et al. (2022) and Koneputugodage et al. (2023). **Chamfer Distance (CD1)** The $L_1$ Chamfer distance is based on the two-ways nearest neighbor distance:

$$\text{CD}_1 = \frac{1}{2|\mathcal{S}|} \sum_{v \in \mathcal{S}} \min_{\hat{v} \in \hat{\mathcal{S}}} \|v - \hat{v}\|_2 + \frac{1}{2|\hat{\mathcal{S}}|} \sum_{\hat{v} \in \hat{\mathcal{S}}} \min_{v \in \mathcal{S}} \|\hat{v} - v\|_2.$$

**Chamfer Distance (CD2)** The $L_2$ Chamfer distance is based on the two-ways nearest neighbor squared distance:

$$\text{CD}_2 = \frac{1}{2|\mathcal{S}|} \sum_{v \in \mathcal{S}} \min_{\hat{v} \in \hat{\mathcal{S}}} \|v - \hat{v}\|_2^2 + \frac{1}{2|\hat{\mathcal{S}}|} \sum_{\hat{v} \in \hat{\mathcal{S}}} \min_{v \in \mathcal{S}} \|\hat{v} - v\|_2^2.$$

**F-Score (FS)** For a given threshold $\tau$, the F-score between the meshes $\mathcal{S}$ and $\hat{\mathcal{S}}$ is defined as:

$$\text{FS}\left(\tau, \mathcal{S}, \hat{\mathcal{S}}\right) = \frac{2\,\text{Recall} \cdot \text{Precision}}{\text{Recall} + \text{Precision}},$$

where

$$\text{Recall}\left(\tau, \mathcal{S}, \hat{\mathcal{S}}\right) = \left| \left\{ v \in \mathcal{S}, \text{ s.t. } \min_{\hat{v} \in \hat{\mathcal{S}}} \|v - \hat{v}\|_2 \langle \tau \right\} \right|,$$
$$\text{Precision}\left(\tau, \mathcal{S}, \hat{\mathcal{S}}\right) = \left| \left\{ \hat{v} \in \hat{\mathcal{S}}, \text{ s.t. } \min_{v \in \mathcal{S}} \|v - \hat{v}\|_2 \langle \tau \right\} \right|.$$

Following Mescheder et al. (2019) and Peng et al. (2020), we set $\tau$ to 0.01.

**Normal consistency (NC)** We denote here by $n_v$ the normal at a point $v$ in $\mathcal{S}$. The normal consistency between two meshes $\mathcal{S}$ and $\hat{\mathcal{S}}$ is defined as:

$$\text{NC} = \frac{1}{2|\mathcal{S}|} \sum_{v \in \mathcal{S}} n_v \cdot n_{\text{closest}(v, \hat{\mathcal{S}})} + \frac{1}{2|\hat{\mathcal{S}}|} \sum_{\hat{v} \in \hat{\mathcal{S}}} n_{\hat{v}} \cdot n_{\text{closest}(\hat{v}, \mathcal{S})},$$

where

$$\text{closest}(v, \hat{\mathcal{S}}) = \text{argmin}_{\hat{v} \in \hat{\mathcal{S}}} \|v - \hat{v}\|_2.$$

