# OpenReview forum: "Distributionally Robust Surface Reconstruction from Sparse Point Clouds"
_ICLR.cc/2025/Conference — ICLR 2025 Conference Withdrawn Submission_

### Official Review · Reviewer_iznP · 2024-10-20

**Soundness:** 3
**Presentation:** 2
**Contribution:** 2
**Rating:** 1
**Confidence:** 3

**Summary:**

The proposed method integrates DRO, which assumes worse cases for SDF learning (WDRO) and further improves its training time and performance through optimal transport (SDRO).

**Strengths:**

1 The proposed method works effectively on sparse, noisy, and unoriented point cloud data, minimizing data quality or density requirements and making it robust to noisy inputs.

2 Unlike prior works using smoothness to prevent overfitting on noise, the work integrates DRO instead. This explores a new direction for the task.

3 The proposed method is benchmarked on several public datasets and real-world datasets.

**Weaknesses:**

1 As the work mentions in the limitation section, the proposed method is sensitive to the hyperparameter. Also, a hyperparameter search needs to be done. Otherwise, the performance is unstable (Figures 9 and 10).

2 Although the mesh reconstruction is much more completed than prior works, I feel it is smoother. However, I am not sure since no ground truth mesh is presented. Please consider including ground truth meshes.

3 Minor writing issue:
- Line 167 uses "Fig" while other lines use "Figure."
- There is an extra "the" in line 77.
- Line 475-476, the "Fig. " miss number.

**Questions:**

1 Since DRO assumes the worst case, I wonder whether it causes inaccuracies in the SDF when the worst case significantly deviates from the original distribution. Conversely, if the worst case only slightly deviates from the original distribution, DRO might have little effect. How does the method handle this situation?

2 It looks like lambda can handle some part of the issue I mentioned in 1 above. However, lambda depends on the ball radius in the Danskin theorem to grow or shrink. How can the radius be accurately defined to prevent the issue in 1?

3 Can the proposed method handle partial/occluded point cloud?

**I thank the other reviewers for pointing out the similarity issue with [1]. Currently, I am not convinced by the author's excuse: 'Some similarities to NAP resulted from the rushed, last-minute nature of the submission,' as mentioned in the global comment. I am quite disappointed by this. Regardless of the quality of the work, ethical issues should be considered and addressed first. I will reconsider this work only if there is no evidence of plagiarism. I now question whether some parts of the paper were written by an LLM, which might explain why the authors did not notice the similarity issue.**

[1]  Ouasfi, Amine, and Adnane Boukhayma. "Few-shot unsupervised implicit neural shape representation learning with spatial adversaries." ICML 2024.

**Details Of Ethics Concerns:**

I thank the other reviewers for pointing out the similarity issue with [1]. Currently, I am not convinced by the author's excuse: 'Some similarities to NAP resulted from the rushed, last-minute nature of the submission,' as mentioned in the global comment. I am quite disappointed by this. Regardless of the quality of the work, ethical issues should be considered and addressed first. I will reconsider this work only if there is no evidence of plagiarism. I now question whether some parts of the paper were written by an LLM, which might explain why the authors did not notice the similarity issue.

[1]  Ouasfi, Amine, and Adnane Boukhayma. "Few-shot unsupervised implicit neural shape representation learning with spatial adversaries." ICML 2024.

---

### Official Review · Reviewer_vk9U · 2024-11-01

**Soundness:** 1
**Presentation:** 1
**Contribution:** 1
**Rating:** 1
**Confidence:** 5

**Summary:**

Unlike existing methods that rely on smoothness priors, this approach is based on a distributionally robust optimization (DRO) framework, incorporating a unique regularization term that leverages samples from uncertainty regions around the model to enhance SDF learning. By employing a tractable dual formulation, the method achieves stable and efficient SDF optimization. However, the proposed method is **extremely similar** to method [1], although method [1] is cited in the manuscript. And the **formulas** used in the manuscript are **hard** to follow and understand, especially for those without basic knowledge of DRO. Some sentences are even the **same** as [1], violating academic ethics.. Besides, the **experiment results **are problematic, making its **effectiveness** questionable.
[1] Ouasfi, Amine, and Adnane Boukhayma. "Few-shot unsupervised implicit neural shape representation learning with spatial adversaries." ICML 2024.

**Strengths:**

Utilize the distributionally robust optimization framework to optimize the SDFs from sparse point clouds.

**Weaknesses:**

1) **The proposed method and the writing are extremely similar to method [1].** Here are some similar and even the same sentences.
* 'Within the realm of 3D shape representation, Neural Signed Distance Functions (SDF) have demonstrated remarkable potential in faithfully encoding intricate shape geometry' in the abstract is the same as [1].
* 'Nevertheless, the process of learning SDFs from sparse 3D point clouds without actual supervision continues to be a highly challenging task' in the abstract is modified from 'However, learning SDFs from sparse 3D point clouds in the absence of ground truth supervision remains a very challenging task.' in [1].
* Figure 1 is the same as that in [1], only the color is changed.
* Many sentences in Sec. 2 are copied from [1].
* Eq. (1)-(4) are the same as [1], only the letters are chenged.
* Some sentences in Algorithm 1 are the same as that in [1].
* Sec 4.1, 4.2 , 4.3 and 4.5 are totally the same as that in [1], withour changing any words.
* The captions of all the figures and tables are the same as thoes in [1].
2) **The formulas are similar or even the same as [1].** And Eqs. (6)-(10) are hard to follow and undersand. So what is the motivation of them? What is their relationship with surface reconstruction from point cloud?
3) **The experimental results are problematic.**
* Some recent methods are not compared, such as OSP [2] and CAP-UDF [3]
* The baseline methods do not consists of method [1].
* There is no efficiency analysis about running time and GPU memory.


[2] Ma, Baorui, Yu-Shen Liu, and Zhizhong Han. "Reconstructing surfaces for sparse point clouds with on-surface priors." Proceedings of the IEEE/CVF Conference on Computer Vision and Pattern Recognition. 2022.
[3] Zhou, Junsheng, et al. "CAP-UDF: Learning Unsigned Distance Functions Progressively from Raw Point Clouds with Consistency-Aware Field Optimization." IEEE Transactions on Pattern Analysis and Machine Intelligence (2024).

**Questions:**

1) What is the difference from method [1]?
2) What about the comparison to those UDF-based methods?
3) Can the proposed method deal with more general shapes, such as those with open boundaries.

---

### Official Review · Reviewer_pjeb · 2024-11-01

**Soundness:** 3
**Presentation:** 2
**Contribution:** 2
**Rating:** 3
**Confidence:** 5

**Summary:**

The paper tackles the neural implicit fitting problem. Based on the previous paper Neural-Pull, the author points out that the current neural implicit methods tend to overfit especially when input point clouds are sparse. To tackle the problem, the author proposes instead of using a fixed distribution to sample the training points, the different sample distributions should be taken into account, and the characteristic of the sample distribution into the loss function. Thus, the author proposes to use a "worst-case sample" to make the model more robust and in order to measure the "worst-case" sample, the author proposes to use Wasserstein distance to indicate the sample "quality". Then add it to the original loss of NP. To be able to optimize it, the author proposes to turn it into a dual reformation, which essentially just adds a dual parameter according to the sample distribution. To solve the slow convergence problem, the author proposes to add entropic regularisation.

**Update**
I read the other reviewers opinion and I agree that the paper is highly similar to
[1] Ouasfi, Amine, and Adnane Boukhayma. "Few-shot unsupervised implicit neural shape representation learning with spatial adversaries." ICML 2024.

**Strengths:**

- 1. I like the idea of adding distribution properties into account when sampling the training points.
- 2. reformulating the problem into dual form and adding entropic regularization are novel theoretical contributions.
- 3. qualitative experiments show that the method performs well.

**Weaknesses:**

I divide the weakness into two parts. Presentation-wise and method-wise.

**Method-wise**
- The method is built on Neural-Pull and improved mainly by sampling multiple times and considering the sampled points distribution, but the ultimate goal is the work on sparse and noisy point clouds. Would Neural-Pull be a reasonable foundation for this task? In my personal experience with NP, the biggest problem when the point cloud is too sparse is that the nearest neighbor search fails to find the correct point to pull. Another problem is when there is a planar surface, the NP cannot reconstruct it, I assume it is because the normal direction is the same for all points on a planar. ( well, gladly I saw in Fig 4. that the proposed method can overcome this issue.)
- As the paper claims to work on sparse and noise point clouds, I would like to see more comparisons of different density point clouds' performance. Now I only saw results point clouds with 1024 points. Maybe it is worth showing more variety of density point clouds and how the proposed methods compare to previous methods.
From this paper: Enhancing Surface Neural Implicits with Curvature-Guided Sampling and Uncertainty-Augmented Representations. They could obtain good results from around 5k points. Maybe the author can use 1024/4096/8196/16384 number of points? I would expect all methods to obtain good results on 16384 points.
- I also would like to see the qualitative results of different noise level inputs. Maybe also visualize the noise point clouds to see the quality of the input. If 0.025 is noise enough. I would suggest the author explore the limit of the noise the method can handle, like adding very large noise on the input.
- The method claims to work on a sparse input point cloud, but I can imagine the performance also relies on the distribution of the input points, namely, the sparse points should be relatively uniformly distributed over the surface. Otherwise, if the points are gathered in one location and hardly occupied on other surface areas, would the method still work?
- There is another baseline worth comparing with DiffCD: A Symmetric Differentiable Chamfer Distance for Neural Implicit Surface Fitting [ECCV24].


**Presentation-wise**
- All the figures that show the qualitative result of the reconstructed meshes, are not well visible. (figure 4 is fine, others are way too small.) I noticed that page 10 is relatively empty, the author might increase the size of figures 2,3,5,6. Especially Fig. 5, 6 show the method obtained good results on real-world data.
- Maybe re-organize the introduction and related work part. Now a lot of related work has been mentioned in the introduction mixed with the paper's initiation, approaches, and the problem the author tried to solve. That makes the introduction very long and related work too short. the author could make the introduction focus on motivating the problem and briefly outline the proposed approach, followed by a separate, more comprehensive related work section.
- Please change the subset E of $\mathbb{R}^3$ to some other math font, it loos like a `[1]` rotated 90 deg. Same as $Q$ in eq (1). looks a bit weird.

**Questions:**

- the author mentioned the Video in the supplementary (Line 66).  I don't see any separate supplementary to download.
- what if the input points are not uniformly sampled on the surface? The author tried a quite extreme case that only 1024 points to recover the surface, but I assume the points are uniformly sampled? How about some missing areas where no samples exist? Can the method still give relatively reasonable results?

I agree with the other reviewers that the paper has too much similarity with [1] Ouasfi, Amine, and Adnane Boukhayma. "Few-shot unsupervised implicit neural shape representation learning with spatial adversaries." ICML 2024.
Therefore I would change my rating until the author gives a reasonable explanation.

---

### Official Review · Reviewer_byED · 2024-11-02

**Soundness:** 3
**Presentation:** 2
**Contribution:** 2
**Rating:** 5
**Confidence:** 4

**Summary:**

This paper utilizes a distributionally robust optimization (DRO) framework with a regularization term to improve the learned SDFs for unsupervised surface reconstruction from sparse point clouds, which is different from most previous methods relying on smoothness priors.

**Strengths:**

The utilization of DRO makes the framework benefit from a stable and efficient optimization of SDFs. Experiments show the proposed method is effective on the selected dataset.

**Weaknesses:**

The main issues are about weak novelty and inadequate validation. The submission is very similar to the recent paper [1] by Ouasfi et al, in terms of strategy and presentation, which weakens novelty. The validation in inadequate, as explain in the question sessions in detail. Also there are many writing errors throughou the paper.  I could change my rating, depending on how my questions and concerns are addressed by the rebuttal.

 (  [1] Ouasfi, Amine, and Adnane Boukhayma. "Few-shot unsupervised implicit neural shape representation learning with spatial adversaries." ICML 2024. )

The proposed method is similar to the method [1] above, differing only in a few formulas and additional experiments. Some text is nearly identical to that in [1], such as the phrase in the abstract: “Within the realm of 3D shape representation, Neural Signed Distance Functions (SDF) have demonstrated remarkable potential in faithfully encoding intricate shape geometry,” which is copied verbatim from [1].  The authors need to carefully explain the similarity and differences with [1].

**Questions:**

First, I list two references to be referred to below:

   [1] Ouasfi, Amine, and Adnane Boukhayma. "Few-shot unsupervised implicit neural shape representation learning with spatial adversaries." ICML 2024.
   [2] Ma, Baorui, Yu-Shen Liu, and Zhizhong Han. "Reconstructing surfaces for sparse point clouds with on-surface priors." Proceedings of the IEEE/CVF Conference on Computer Vision and Pattern Recognition. 2022

About novelty:

(1)	The proposed method is similar to the method [1] above, differing only in a few formulas and additional experiments. Some text is nearly identical to that in [1], such as the phrase in the abstract: “Within the realm of 3D shape representation, Neural Signed Distance Functions (SDF) have demonstrated remarkable potential in faithfully encoding intricate shape geometry,” which is copied verbatim from [1].  The authors need to carefully explain the similarity and differences with [1].

About literature review:

(2)	The main innovation of the paper lies in the use of DRO; however, the related work section does not discuss recent developments in DRO research. Please add  more discussions on recent works in DRO?

About Validation and Comparison:

(3)	The baselines used for the Eikonal approach are DIGS and OG-INR, but the results for these two are not the SOTA. The paper mentions StEik and Neural-Singular-Hessian, both of which outperform DIGS and OG-INR in reconstruction quality. The paper does not include a comparison with them. More comparison is needed?

(4)	The baseline methods do not include method in paper [1]  listed above or some recent methods like OPS in paper [2] listed above. Additionally, some baseline results in Table 1 differ from those reported in method [1], casting doubt on the reliability of this paper’s experimental results. Please explain.  Please explain?

(5)	The paper only discusses Neural-Pull. It is not clear whether the sampling strategy improvement using DRO is generalizable and can be applied to other methods, such as Eikonal -based methods like SIREN, DIGS, and StEik. Please discuss?

(6)	The training time comparison experiment provided in the paper shows the changes in reconstruction quality for Neural-Pull, SDRO, and WDRO (under different training times) but does not offer efficiency comparisons with other baselines. Please add experoments?

(7)	 The quantitative comparison given in the supplementary material on dense point clouds indicates that the proposed method performs worse than the baselines on dense point clouds. The paper does not specify the scale of these dense point clouds, and the other experiments are conducted on the same sparse point cloud scale, making it difficult to determine the effectiveness of the proposed method on varying scales of sparse point cloud data. Please add validation?

(8)	Some tables are unclear in presentation; for example, Table 5 provides numerical comparisons, but neither the table nor the main text explains which metric is being compared. There are also some noticeable writing errors in the text. Please clarify?

**Details Of Ethics Concerns:**

Suspicious ethic issue:  The authors need to explain why some text is nearly identical to that in [1], such as the phrase in the abstract: “Within the realm of 3D shape representation, Neural Signed Distance Functions (SDF) have demonstrated remarkable potential in faithfully encoding intricate shape geometry,” which is copied verbatim from [1].

---

### Official Review · Reviewer_5DiT · 2024-11-03

**Soundness:** 2
**Presentation:** 2
**Contribution:** 3
**Rating:** 5
**Confidence:** 3

**Summary:**

This work develops an implicit SDF reconstruction approach for sparse point clouds with application of distributionally robust optimisation. Taking Neural-Pull as the baseline, where the query points to the SDF are sampled from a fixed Gaussian distribution, the approach joinly learns another “worst-case” distribution. The expectation is that such an adversarial scenario will benefit learning the SDF of an input point cloud to a large degree of accuracy, especially if it’s sparse and/or noisy. The work develops two variants of the DRO-based reconstruction, which incur a surrogate distribution for the query points and is assumed to embed an adversarial nature of the nominal one.
The experiments indicate that both variants are on par with or outperform previous work on the accuracy metrics (e.g. Chamfer distance) across three benchmarks. The qualitative results show that DRO allows the reconstruction to recover finer details in the 3D shape as well as to get rid of outliers in the input point cloud.

**Strengths:**

* It is an exciting new idea to combine distributionally robust optimisation with implicit SDF reconstruction.
* The empirical results on reconstructing from sparse and noisy point clouds are impressive. The choice of benchmarks is quite broad and laudable in scope.
* I like the abundance of the qualitative results (though would prefer specificity in the main paper and leave the broader comparison for the supplemental material).
* The experiments also include the analysis of the runtime of the two variants of the the proposed DRO objective.

**Weaknesses:**

* I like that the main hypothesis, the potential for a better sampling strategy, is backed by previous work. However, the link is not entirely clear to me. After all, the provided references link to learning-based methods. Here, by contrast, we deal with an optimisation problem, where the network approximates a specific SDF and does not have any generalisation capacity. Fig. 1 makes an earnest effort to illustrate the idea, but I still fail to see why simulating a worst-case scenario for sampling could have benefits for an optimisation (not learning) problem. I hope to see a more analytical argument for why DRO is expected to work in this setting.
* The writing and especially the presentation of the results could be improved. I would have preferred more targeted, descriptive and higher-resolution qualitative examples as opposed to thumbnail grids (e.g. Fig. 2,3,5,6).
* The approach works well for noisy and sparse point clouds. However, it should operate around the same to Neural-Pull (the baseline) in the noiseless setting. Surprisingly, it works even better than NP, as shown in Tab. 4. The loss, as developed in Eq. (5) and (10), can affect the convergence speed, but why it should improve the noise-free reconstruction is unclear.

Minor remarks:
* The use of \lambda is overloaded (e.g. \lambda and \lambda_1, \lambda_2 have quite different meanings)
* Some typos (e.g. l. 077, l.361)

**Questions:**

* Is there a geometric interpretation of the DRO-based approach? Why should sampling the queries from a different Q’ improve the reconstruction?
* Would it be possible to visualise the distributions Q and Q’/Q_{x,\rho} for at least some toy / 2D example? I'd be curious to see an ablation experiment interpolating from Q' to Q to observe that the improvement in the reconstruction quality really comes from Q'.
* Why the specific input size of 1024 points (l. 317)? How would the method scale in comparison to the baseline if the number of points is increased or decreased?
* Qualitatively, I observe that NP and SparseOcc may have off-surface elements related to the noise in the point cloud (e.g. second to last row in Fig. 6). The proposed approach does not have them. Is there any post-processing, and, if not, how exactly does it circumvent this problem?

---

### Author Response · Authors · 2024-11-12

First, we thank all reviewers for their time and valuable feedback

## Further Positioning wrt NAP [1]

Although we duly introduced NAP[1] in L054, and mentioned the position w.r.t. it in lines L074 and L088, we apologize if this positioning was not extensive enough. We will elaborate on it in the next version, as outlined in the following:
As introduced in L054 and L074, NAP[1] focuses on how training samples are generated and understanding to which extent training distributions affect performance. It uses pointwise adversarial samples to regularize the learning process.  **This is a special case of distributionally robust optimization with Wasserstein uncertainty sets**. In this paper we rely on tractable dual formulations (Blanchet & Murthy, 2019, Sinha et al., 2017) of the DRO problem with Wasserstein distribution metric. Contrary to NAP, WDRO (Equation 5) relies on a soft-ball projection controlled by the parameter λ that is adjusted throughout the training, ensuring that it grows when the worst-case sample distance from the initial queries exceeds the Wasserstein ball radius ε  (L277). Following Wang et al. (2021, we show that using **the Sinkhorn distance to define the uncertainty sets** is an effective strategy **under high levels of noise (L475, L484), while NAP and WDRO can be slightly better under low levels of noise**. This is witnessed by the noise ablation (Table 4) as well as in qualitative comparisons on BlendedMVS, Tanks Temples (Figure 6) and SemanticPOSS (Figure 5) datasets where we outperform these baselines.


## Similarities to NAP [1]:

We would like to first remind that in our original manuscript:
- we have cited and introduced NAP in lines  L054,L088 and commented on its performance in lines L475, L484.
- We have compared qualitatively to NAP in figures 2, 3, 5,6.
- We have compared numerically to NAP in Tables 1, 2, 3, 4.

Some similarities to NAP resulted from the rushed last minute nature of the submission, and we sincerely apologize to the reviewers for that. However, this paper and NAP [1] also naturally share many similarities because they address the same problem, building on the same baseline, and using the same evaluation benchmark. **We are currently rewriting the paper to remove all similar phrasing, thus resolving this issue completely in the next version**.

[1]  Ouasfi, Amine, and Boukhayma Adnane. "Few-shot unsupervised implicit neural shape representation learning with spatial adversaries." ICML 2024.

---

### Author Response · Authors · 2024-11-13

After careful consideration, we have decided to withdraw this submission. We thank the reviewers for their valuable feedback and apologize for any inconvenience caused.

---

### Note · Authors · 2024-11-13

I have read and agree with the venue's withdrawal policy on behalf of myself and my co-authors.